# Antiviral Effects of Plant-Derived Essential Oils and Their Components: An Updated Review

**DOI:** 10.3390/molecules25112627

**Published:** 2020-06-05

**Authors:** Li Ma, Lei Yao

**Affiliations:** 1R&D Center for Aromatic Plants, Shanghai Jiao Tong University, Shanghai 200240, China; malimali2006@sjtu.edu.cn; 2Department of Landscape Architecture, Shanghai Jiao Tong University, Shanghai 200240, China

**Keywords:** essential oil, antiviral activity, resistance, mechanism of action

## Abstract

The presence of resistance to available antivirals calls for the development of novel therapeutic agents. Plant-derived essential oils may serve as alternative sources of virus-induced disease therapy. Previous studies have demonstrated essential oils to be excellent candidates to treat antiviral-resistant infection associated with their chemical complexity which confers broad-spectrum mechanisms of action and non-specific antiviral properties. However, almost no comprehensive reviews are updated to generalize knowledge in this regard and disclose the interplay between the components and their antiviral activities. This review provides an up-to-date overview of the antiviral efficacy of essential oils from a wide range of plant species and their characteristic components, as well as their overall mechanisms of action, focusing on the last decade. The roles of individual components relative to the overall antiviral efficacy of essential oils, together with the antiviral activity of essential oils in comparison with commercial drugs are also discussed. Lastly, the inadequacies in current research and future research are put forward. This review will provide references in the design of new drug prototypes and improve our understanding of the proper applications of essential oils in the future.

## 1. Introduction

A virus is a small particle (20–300 nm) containing merely genetic substances wrapped with proteins and, sometimes, lipids [1]. They infect host cells to fulfill self-replication. The viral diseases remain a significant worldwide concern. Currently, few desirable antiviral drugs are available, and, in most cases, they exhibit either intracellular or extracellular antiviral properties. Acyclovir, a nucleoside analogue against human herpes viruses (HSV), blocks viral replication via inhibition on viral DNA polymerases [2]. Amantadine, an M2 ion channel inhibitor for influenza virus (IFV) A, blocks the influx of H^+^ ion into the virus and prevents the virus from uncoating [3]. Zidovudine, a nucleoside analog reverse transcriptase inhibitor for the treatment of human immunodeficiency virus (HIV) infection, interferes with, through its azide moiety, the formation of phosphodiester linkages essential for viral DNA amplification [4]. However, the narrow spectrum of the mechanisms of action would lead to continuous drug-resistance [5], a phenomenon more prevalent in immunocompromised hosts, not to mention the side effects. Therefore, discovery of new therapeutic alternatives with multiple viral targets to avoid resistance comes to the forefront.

Plant extracts play an important role in the folk medicine as therapeutic substitutes to relieve suffering. Hitherto, natural products are still major resources for discovery of new therapeutic agents. Plant derived-essential oils (EOs) are complex volatiles composed of miscellaneous phytochemicals, such as monoterpenes, sesquiterpenes and phenylpropanoids, etc., the structures of which are frequently reviewed elsewhere [6,7]. Essential oils as antibacterials, antifungals, antioxidants, etc., have been extensively investigated in numerous studies for decades [8,9,10]. Virucidal effects of EOs extracted from numerous aromatic and herbal plants are also well documented on a variety of viruses, such as IFV, HSV, HIV, yellow fever virus, and avian influenza, etc. [11,12,13,14]. In a recent study, Cagno et al. [2] reported that the EO from *Salvia desoleana* Atzei & V. Picci significantly suppressed the acyclovir resistant HSV-2 strains with a 50% effective concentration value (IC_50_) of 28.57 μg/mL, far less than that of acyclovir (71.84 μg/mL). In another study, EOs from five plant species (*Zataria multiflora*, *Artemisia kermanensis*, *Eucalyptus caesia*, *Satureja hortensis*, *Rosmarinus officinalis*) also exhibited better anti-HSV activities than acyclovir [15]. These results suggest that EOs can be potential therapeutic agents for the treatment of virus infection as well as prototypes for new antiviral drug selection.

Although a number of studies investigated the antiviral activities of plant-derived EOs in the past decades, they are fairly extensive and not interconnected to form an overall trend. The antimicrobial, antifungal, antioxidant and anti-inflammatory effects are extensively reviewed at times [16,17,18], while the up-to-date antiviral effects are seldom generalized. One related book chapter review published ~10 years ago summarized the inhibitory potency of common EOs and EO-derived components and the overall mechanisms of action [19]. However, the antiviral effects of EOs relative to their components and commercial drugs, in particular, are inadequately reviewed and advances in EOs as antivirals are not updated. Another two reviews discussing biological activities of EOs (antimicrobial, antifungal and antiviral activities) only allow limited space for antivirals [1,20]. Therefore, in this review, we focus on in vitro antiviral studies published within 10 years, with the purpose to provide up-to-date information on the antiviral properties of EOs. Animal-borne viruses are covered herein while phytoviruses are out the scope of the article.

## 2. Mechanisms of Action

### 2.1. Time-of-Addition Experiment

Viruses attach, penetrate, and enter the host cell, where genetic substances are replicated, followed by the creation and release of new virions [19]. Adverse action on related targets involved in the infection phase will inhibit the viral infectivity (Figure 1). Mechanisms of action of EOs (including their components) against viruses are usually determined via the manipulation of time-of-addition assays. Cultured cells are pretreated with the EOs for 1 h prior to virus infection (pre-viral infection). A negative result indicates that EOs do not affect viral attachment by blocking host cell receptors. Alternatively, viruses are pretreated with EOs for 1 h followed by incubation simultaneously with host cells (simultaneous viral infection). A positive result indicates that EOs interfere with free virions by modifying the virus envelope structure or masking the viral proteins, which are necessary for viral adsorption and entry into the host cells. Otherwise, EOs are added to the infected cells at different time intervals of the viral infection lifecycle (from penetration to progeny production) (post viral infection). In this case, the stage of viral infection cycle at which EOs act against viruses can be accessed. So far, the time-of-addition assay is the most widely used procedure to investigate the overall intercellular and intracellular inhibitory properties of EOs. Collectively, EOs and their components majorly act on free viruses directly (intercellular mode of action). Multiple modes of action also exist, which may be EO-dependent.

### 2.2. Morphological Alteration

Relying on one assay only nonspecific information on mechanisms of action is derived. For example, the time-of-addition experiment cannot tell whether the EOs influence viral adsorption by destroying or masking the virus. Fortunately, transmission electron microscopy (TEM) imaging was used to visualize structural changes of the virus. With the aid of TEM, Gilling et al. [21] found that murine norovirus (a non-enveloped virus) exposed to 4% oregano EO expanded in size from 20–35 nm to 40–75 nm in diameter but appeared intact in morphology. Meanwhile, 0.5% carvacrol treated murine norovirus expanded from normal to ~900 nm in diameter, resulting in capsid disintegration. The capsid in the non-enveloped virus protects the viral RNA from disintegration and triggers infection by adsorption to host cells [22]. Nevertheless, murine norovirus whose capsid was partially degraded may still be infectious since the RNase I protection assay revealed that capsid degradation did not lead to noticeable viral RNA reduction. The results implied that the carvacrol interfered with virus adsorption to host cells via binding to or masking the capsid but not via structural damage to the virus.

### 2.3. Protein Inhibition

Hemagglutinin (HA), an important membrane protein of the IFV, allows the virus to enter and exit the host cell [23]. HA can cause the agglutination of red blood cells, so hemagglutination inhibition assay is usually used to test the effect of EOs on viral adsorption to host cells. Absence of agglutination indicates inhibition of HA activity. Neuraminidase (NA) is another important surface protein for initial IFV infection, which is also a target of the synthetic drugs, zanamivir and oseltamivir. Summing up, EOs tend to act on HA more than on NA, which is dependent on types of the EO or compound. As an example, cedar leaf EO was found to strongly inhibit HA of IFV, while thujone and α-pinene, two major components of the EO, did not [24], for which specific underlying mechanisms are unclear.

Tat (trans-activator of transcription) is an essential protein for HIV transcription. Tat interacts with TAR (trans-activation-responsive region) RNA, which is required for HIV-1 replication. Therefore, Tat/TAR-RNA complex could be a target of HIV-1 inhibitors. Feriotto et al. [25] examined the EO-treated Tat/TAR-RNA complex using gel electrophoresis and found that EOs of *Thymus vulgaris*, *Cymbopogon citratus*, and *Rosmarinus officinalis* interacted directly with Tat protein and destabilized Tat/TAR-RNA complex.

In an in silico study, Pajaro-Castro et al. [13] calculated the affinity scores of EO-derived components to Dengue virus proteins and then identified the interactions between the highly scored components and the viral proteins. The results showed that most of the components inhibited the Dengue virus via binding to the nonpolar domain of the proteins through hydrophobic interactions. In silico studies conducted by Sharma and kaur [26] showed that 1,8-cineole from eucalyptus EO effectively bound to COVID-19 proteinase via hydrophobic interactions, hydrogen bond and ionic interactions. Isothymol from the EO of *Ammoides verticillata* (Desf.) Briq. was reported to be a good inhibitor against angiotensin converting enzyme 2, a receptor of COVID-19, via Pi-H bonding [27]. However, whether these interactions apply to cases with other viruses and EOs involved and whether the polarity of the EO components affects the antiviral efficacy remain to be elucidated.

### 2.4. Other Mechanisms of Action

During the infection process, the IFV uncoating step needs an acidic endosomal and lysosomal environment [28]. An investigation of cellular endosomal/lysosomal pH alteration after EO application can track the effect of EOs on early stages of the viral replication lifecycle. For example, Garozzo et al. [29] reported that tea tree EO and its major component terpinen-4-ol inhibited viral replication by way of interrupting the acidification of intralysosomal compartment which is essential for virus uncoating. It is known that viral infection induces intracellular glutathione depletion in accompany with the redox change [30]. EO-derived components, piperitenone oxide for instance, could inhibit the late stage of HSV-1 lifecycle by targeting on redox signaling pathway [30]. Also, EOs may target genome related sites as revealed by genetic approaches [31]. Towards a certain type of virus, mechanisms of action may be EO dependent [32], possibly owing to the constitutional variations of the EOs. Summing up, the direct interaction of EOs with free viruses tends to be the most common mode of action.

## 3. In Vitro Studies of Antiviral Activities of Essential Oils

Antiviral activities are generally evaluated in vitro via cytopathic effect reduction assay, plaque reduction assay and viral yield reduction assay [20]. Generally, all antivirals inhibit the viruses in a dose-dependent manner. To ensure that the EOs at the assayed concentrations do not exert toxicity on the host cells, we must test the cytotoxicity of EOs, which is described in terms of CC_50_ (50% cytotoxic concentration), corresponding to the EO concentration that reduces the cell viability by 50%. Viral activity is evaluated by the value of IC_50_ (50% infection concentration), i.e., the EO concentration required to reduce viral infection by 50%. The antiviral selectivity index (SI), a measure of the therapeutic suitability of the EO, is calculated as the ratio of CC_50_ to IC_50_ [33]. To define anti-infective potential in natural products, IC_50_ values should be below a threshold of 100 μg/mL for mixtures and 25 μM for pure compounds [34]. An SI value is >4 is deemed acceptable [35]. The antiviral properties of EOs from different aromatic plants and EO-derived components on different virus are summarized in Table 1 and Table 2.

### 3.1. Human Herpes Virus

Human herpes viruses (e.g., HSV-1 and HSV-2), enveloped DNA viruses, have received the most attention in the last decade. Essential oils from Star Anise, Australian tea tree, oregano, *Eucalyptus caesia*, to name a few, have been demonstrated to exhibit high anti-HSV-1 activities in vitro (Table 1). Table 1 indicates that Star Anise EO is the most potent with an IC_50_ value of 1 μg/mL and a SI value of 160 [36], far away from the recommended cutoff (IC_50_ < 100 μg/mL; SI > 4). Moreover, *Mentha suaveolens* and Australian tea tree are also potential efficient antivirals [30]. Antiviral screening studies using EOs from *Umbelliferae*, *Labiatae*, *Myrtaceae*, *Lamiaceae* plants indicated that HSV-1 was susceptible to a wide range of aromatic plants [15,36]. With regard to the antiviral efficacy of EO-derived components, β-caryophyllene stands out with IC_50_ and SI values of 0.25 μg/mL (equal to 1.2 μM) and 140 (Table 2), respectively. This indicates that β-caryophyllene-containing plants, such as black pepper, cannabis, cinnamon, oregano and cloves, may serve as potential sources for selection of HSV antivirals [37]. In a study, a wide range of common components in EOs have been screened for their anti-herpes activities and all exhibited high antiviral activities at the concentration range of 0.025–0.8 μg/mL with the exception of L-bornyl acetate and D-limonene [36]. Elsewhere, 1,8-cinole [38], eugenol [39], and *p*-cymene [14] were pointed out in a few studies to be inappropriate for the treatment of HSV-1 due to inefficacy (IC_50_ > 25 μM) or high cytotoxicity (SI < 4).

### 3.2. Influenza Virus

Influenza viruses are enveloped RNA viruses. Essential oils and their constituents were frequently examined in vitro and reported to exhibit high therapeutic potential in a dose-dependent manner. Choi [54] screened the anti-influenza A/WS/33 virus activities of 62 EOs and found that, among them, marjoram, clary sage, and anise oils exhibited higher efficacy (IC_50_ < 100 μg/mL) than oseltamivir. Elsewhere, EOs from *Thymus vulgaris*, *Cinnamomum zeylanicum*, *Citrus bergamia*, etc., were also reported to show high antiviral efficacy [23] (Table 1). As for EO-derived components, germacrone is among the most effective ones against IFV with a SI value larger than 41, which may be due to its broad-spectrum of mechanisms of action by inhibiting multiple steps of viral replication [53]. Also, germacrone was shown to effectively inhibit multiple strains of feline caliciviruses, non-enveloped RNA viruses [55]. These results suggested that germacrone, a major component from rhizoma curcuma, a traditional Chinese herb, could be used as a promising broad-spectrum therapeutic agent or a reference for novel antiviral drug development.

Interestingly, some components such as 1,8-cinole and eugenol, which are ineffective against HSV-1, was found to exhibit high anti-influenza activities [23,56]. Additionally, other oxygen-bearing components such as β-santalol [52] and terpinen-4-ol [29] were also reported to be major bioactive components against IFV. This contrasting result implies that the anti-HSV and anti-influenza effects of some components may be complementary. There seems to be a trend that non-oxygenated terpene hydrocarbons are more effective to HSV and oxygenated terpenes to IFV, a hypothesis which needs to be tested.

### 3.3. Non-Enveloped Viruses

Antiviral studies in the past years focused on enveloped viruses, while non-enveloped viruses received relatively less attention. This is not surprising since a large portion of EOs were reported to exert high antiviral activity prior to host cell infection via interaction with the envelope, while non-enveloped viruses have fewer active sites that the EOs are able to target. The EO from *Osmunda regalis*, a Tunisian fern, demonstrated its superior anti-Coxsackie activity with a IC_50_ value of 2.24 μg/mL and SI value 789.84 [43], far more beyond the threshold. Moreover, *Eucalyptus bicostata* and *Dysphania ambrosioides* EOs also exhibited high anti-Coxsackie activities [42,44]. The oregano EO and its primary component carvacrol were able to strongly inhibit human rotavirus and murine norovirus [21,33]. These studies indicated potential antiviral activities of EOs against non-enveloped viruses. Unfortunately, information was not available regarding detailed mechanisms of action and the bioactive compounds in EO mixture responsible for the antiviral efficacy, further hindering the exploration of their antiviral value.

### 3.4. Other Viruses

*Cymbopogon nardus* EO was reported to inhibit HIV-1 reverse transcriptase activity with a IC_50_ value of 1.2 mg/mL (equal to 1200 μg/mL) and the author postulated that (S)-β-citronellol may be one of the bioactive component [31]. *Thymus vulgaris*, *Cymbopogon citratus*, and *Rosmarinus officinalis* EOs were found to effectively inhibit HIV-1 Tat/TAR-RNA interaction with an IC_50_ ranging from 0.05 to 0.83 μg/mL [25]. However, the SI values were all below the cutoff value of 4. Nevertheless, these findings provide alternative choices for development of HIV antivirals. Elsewhere, β-caryophyllene effectively inactivated Dengue-2 virus targeting at intercellular and intracellular sites with IC_50_ and SI value of 22.5 μM and 71.1 [13], respectively. More recently, EO-derived components, such as 1,8-cineole [26] and isothymol [27], were suggested to be potential inhibitors of COVID-19. Antiviral effects of EOs and their components were also observed on bovine viral diarrhoea virus [33], respiratory syncytial virus [24], yellow fever virus [46], caprine alphaherpesvirus [57], and Zika virus [47], etc. The antiviral potential of EOs and their components on non-enveloped viruses is worthy of further exploration. One thing worth noting is that, although the aromatic compound carvacrol showed a broad-spectrum antiviral effect (Table 2), caution must be taken before application since carvacrol exhibits a high cytotoxicity, as reflected by the relatively low SI values.

## 4. Effect of Polarity of the Components on Anti-Viral Activities

Venturi et al. [58] found that exposure of *Glechon spathulata* and *Glechon marifolia* EOs to light (for 20 d) compromised their anti-HSV-1 efficacy, with a concomitant remarkable decrease in terpenes, especially non-oxygenated monoterpene hydrocarbons, which were postulated to be the major active antivirals. Similarly, a bioassay-guided fractionation of the EO from *Salvia desoleana* suggested that it is the terpene hydrocarbons (Fraction 1) other than the oxygenated monoterpenes (Fraction 2), with germacrene D (54%) and β-caryophyllene (4.8%), the major component of the former fraction, and linalyl acetate (39.7%), α-terpinyl acetate (30.1%), 1,8-cineole (12.6%), and linalool (8.1%), the later, accounting for the anti-HSV-2 activity [2]. Further antiviral test with individual linalyl acetate, α-terpinyl acetate, 1,8-cineole, and linalool demonstrated that these oxygenated-monoterpenes had no antiviral effect on HSV-2 at all, while the non-oxygenated fraction (Fraction 1) showed higher antiviral efficacy (IC_50_ 10.7 μg/mL for Fraction 1) than the EO mixture (IC_50_ 23.7 μg/mL). Elsewhere, Astani et al. [39] compared the anti-HSV-1 activities of six EO-derived phenylpropanoids and sesquiterpene and found that addition of an epoxide or hydroxyl function into the sesquiterpene hydrocarbons incurred decrease in their antiviral activity. These results suggest that increasing polarity (addition of oxygen) likely decreased the anti-HSV activity. On the contrary, Lai et al. [49] investigated the anti-HSV-1 effect of thymol-related monoterpenoids and found that the anti-HSV-1 effect decreased as the polarity of the substituents declines in the order: ˗OH > ˗NH_2_ > ˗CH_3_ > ˗H. The inconsistence remains unclear.

Ralambondrainy et al. [32] reported that monoterpene-rich *Cymbopogon citratus* and *Pelargonium graveolens* EOs exhibited higher antiviral effect on the Ross River Virus, while the sesquiterpene-rich *Vetiveria zizanioides* EO showed an insignificant effect. However, it seems arbitrary for the author to draw an inference that the antiviral competency was ascribed to the monoterpenes without further experimental proofs. Pajaro-Castro et al. [13] calculated and compared the binding affinities of a variety of EO-derived components to structural proteins of Dengue viruses, including monoterpenes and sesquiterpenes, with or without oxygen-bearing functional groups, and found that the sequiterpenes hydrocarbons and some of their oxygenated counterparts, such as α-copaene, germacrene D, β-caryophyllene, and caryophyllene oxide, exhibited the highest affinity. According to Pajaro-Castro et al. [13], the EO components bound to dengue virus proteins majorly via hydrophobic force, from which it is extrapolated that nonpolar terpenes tend to show great interaction with viral proteins. Here, we hypothesize that the binding force of the components to viral surface likely influences the anti-HSV effectiveness of the EO-derived components and the polarity of the components is one influencing factor of the binding affinity. However, limited information is available concerning the effect of polarity (or addition of oxygen) on antiviral efficacy. Germacrone and β-caryophyllene, two characteristic components as we stated above, showed a relatively broad spectrum of antiviral activities. The antiviral capacities of their oxidized form, i.e., germacrene and caryophyllene oxide, however, are almost not addressed, making the evaluation of the roles of oxygen addition to the precursor (non-oxygenated terpene hydrocarbons) tough. Therefore, extensive and systematic research with diverse viruses and EO-components needs to be attempted to make a relatively confirmatory inference.

## 5. Activities of EOs Compared to That of Their Principal Components

It is generally accepted that the EO chemistry determines its bioactivities. Still, there are conflicts regarding the role that individual component plays in the overall EO antiviral activity. *Eucalyptus globulus* and *Salvia officinalis* both contain the major component 1,8-cineole, however, the former was reported to have strong anti-H1N1 activity (IC_50_ < 3.1 μg/mL) while the later was not [23], suggesting that other minor components may be more bioactive than the primary component. Eugenol, a principle component of *Cinnamomum zeylanicum* EO, was found to be as potent as the EO in treatment of H1N1 [23], indicating that the antiviral efficacy of the EO could be ascribed to its principle component. Pilau et al. [33] found that Mexican oregano EO was more efficient than its principle component, carvacrol, against HSV-1, bovine viral diarrhoea virus, and respiratory syncytial virus, while the carvacrol was able to efficiently inhibit rotavirus on which the Mexican oregano EO had no inhibitory effect at all at the tested concentration (25–3200 µg/mL). These results suggest that individual terpene in EO may not contribute equally to the antiviral efficacy of the EO mixture. Therefore, the antiviral efficacy of individual compounds relative to the EO mixture should be case dependent.

Indeed, in some cases, the EO mixture may provide higher SI and a lower toxicity than their isolated single component [33,38], making the EO blend stand out. But this does mean that the EO used as mixtures is always superior and preferable to its single isolate. There exist bioactive components, either minor or primary, that are responsible for EO bioactivity. From a new drug discovery point of view, isolation and in-detail studies of individual EO components that are far more bioactive than the EO mixture deserve our more attention.

## 6. Effects of Essential Oils Relative to Available Drugs

Essential oils may be more potent and versatile than commercial drugs. For example, carvacrol and oregano EOs were able to efficiently inhibit acyclovir-resistant HSV-1 [33]. *Salvia desoleana* EO efficiently suppressed acyclovir-sensitive and acyclovir-resistant HSV-2 strains. Also, the author demonstrated, via chromatographic fractionation of the EO, that the primary isolate, germacrene D, likely accounted for the overall antiviral effect [2]. Similarly, Liao et al. [53] reported that germacrone showed more inhibitory efficacy than ribavirin against multiple IFV strains (including amantadine resistant strains) since it interferes with viral attachment via direct interaction with either the viruses or host cells as well as inhibits early phase of viral lifecycle by impairing viral protein expression and RNA transcription. This suggests that the EO-derived components may avoid targets of M2 ion channel inhibitors (amantadine and rimantadine) and neuraminidase inhibitors (zanamivir and oseltamivir) or exhibit multiple mechanisms of action to override the drug-resistance. It is believed that the compositional complexity of EOs confers co-occurrence of multiple bioactive components and synergism, which tends to render the antiviral competency of EOs more diversified, in contrast to that of the synthetic drugs, which is virus specific.

Current studies demonstrated synergistic or additive activities when the commercial drugs are used in combination with EOs. Pourghanbari et al. [59] found that the combination of oseltamivir and *Melissa officinalis* EO enhanced the effect of oseltamivir on avian influenza A virus (H9N2). In this study, 0.05 μg/mL of oseltamivir reduced the virus genome copy number to 12,000/μL while the blend of 0.05 μg/mL of oseltamivir and 0.5 mg/mL of the EO reduced the number to near zero. Additionally, *Mentha suaveolens* EO and its principle component, piperitenone oxide, also exhibited synergistic activities against HSV-1 in combination with acyclovir [30]. In the study, the number of viral titres in infected host cells was reduced by ~90% when acyclovir (0.05 μg/mL) was administrated in combination with piperitenone oxide (15 μg/mL) than when acyclovir was used alone. Moreover, germacrone in combination with oseltamivir was reported to exhibit additive anti-influenza viral effect both in vitro and in vivo [53], namely, the antiviral capacity of the combination, in terms of inhibition concentration index [60] or infected mice survival rate, was a sum of that of individual contributions. Collectively, the synergistic activity of a combination of drugs and EOs could be applied in future therapeutic treatment of infectious diseases due to such advantages as higher efficacy, broad-spectrum activity, lower cost, and less drug resistance.

## 7. Concluding Remarks and Future Perspectives

The existing references in the past ten years are rather limited and most of them investigated in vitro the antiviral effect of EOs from various families of plants and EO-derived components. Many studies showed the direct action against cell-free virions, the broad-spectrum antiviral activities and the antiviral versatility of EOs. The antiviral potency was tested in clinical studies [61,62] and some achieved pleasant results. As an example, myrtle vaginal suppository containing 0.5% myrtle EO was effective in patients with HPV infection with an HPV clearance rate of 92.6% compared with that of placebo (68%) [62]. Taken together, EOs hold promise as candidates for prophylactic and therapeutic treatment of virus-induced diseases in future.

So far, mechanisms of action of the antiviral activity of EOs are not yet fully illustrated, which is not conductive to a successful application of EOs. Currently, in most cases, investigators can only tell at which stage of viral infection that the EOs action, while the specific sites and interaction mode involved in each step, for instance, are inadequately addressed. Therefore, in-detail and in-depth investigation of mechanisms of action is the foremost thing in future. To construct a whole picture of modes of action, multiple experimental assays should be performed simultaneously. Apart from TEM imaging, cell-binding experiment, RNase I protection experiment, and other biochemical approaches, strategies such as in silico computational design, should be developed, if applicable.

Additionally, to our knowledge, the EO chemistry determines the bioactivity of the EOs, efforts thus need to be made to further reveal the relationship between EO chemistry and the corresponding bioactivity. As discussed in the previous section, the antiviral effectiveness of EOs can be contributed unequally to the active components, either minor or principle ones, and underlying synergism. In this case, fractionation of EO blend, identification of the bioactive individuals and evaluation of their antiviral effectiveness are fairly essential. Existing studies evaluated the antiviral effect of commercially obtained EO-derived compounds, which may not be fully consistent with their fractionated counterparts from EOs. The effects of chemical properties of stereoisomerism and polarity of the components on their antiviral effectiveness need to be figured out. The preparative high-performance chromatography in terms of thin-layer chromatography, gas chromatography, and liquid chromatography, as well as the state-of-the-art equipment for structural elucidation facilitates the fractionation of EO components and evaluation of their bioactivities. As an example, preparative chromatography-based fractionation combined with bioassay-guided test can be a feasible way for rapid screening and isolation of the bioactive individuals, which is essential for new antiviral drug discovery and development.

Last but not least, references regarding the antiviral effect of EOs were not published in a large quantity in the past decade and the focus of the virus type is enveloped ones, especially HSV-1 and IFV. A few studies have already evidenced the great antiviral potency of some EOs and their components on certain non-enveloped viruses. However, the mechanisms of action at the molecular level are inadequately elucidated and plant sources investigated are relatively limited. Therefore, research in this regard deserves our further attention.

## Figures and Tables

**Figure 1 molecules-25-02627-f001:**
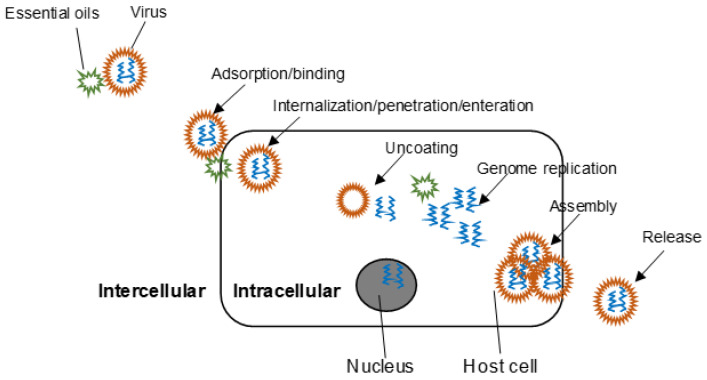
Possible targeting sites of essential oils during viral lifecycle.

**Table 1 molecules-25-02627-t001:** Antiviral activity of different plant-derived essential oils.

No	Source of Essential Oils	Viruses	IC_50_	SI	Intercellular or Intracellular Mechanisms	References
1	Star Anise	HSV-1	1 μg/mL	160	Intercellular	[39]
2	*Mentha suaveolens*	HSV-1	5.1 μg/mL	67	Intracellular	[30]
3	Australian tea tree	HSV-1	13.2 μg/mL	43	Intracellular	[30]
4	*Sinapis arvensis*	HSV-1	0.035%	1.5	Intercellular	[14]
5	*Lallemantia royleana*	HSV-1	0.011%	6.4	Intercellular	[14]
6	*Pulicaria vulgaris*	HSV-1	0.001%	1	Intercellular	[14]
7	Mexican oregano (*Lippia graveolens*)	HSV-1	99.6 μg/mL	7.4	Intercellular	[33]
8	*Zataria multiflora*	HSV-1	0.003%	55.4	ND	[15]
9	*Eucalyptus caesia*	HSV-1	0.007%	38.8	ND	[15]
10	*Artemisia kermanensis*	HSV-1	0.004%	66.4	ND	[15]
11	*Satureja hotensis*	HSV-1	0.008%	32.2	ND	[15]
12	*Rosmarinus officinalis*	HSV-1	0.006%	46.1	ND	[15]
13	*Thymus capitatus*	HSV-1	17.6 μg/mL	6.9	Intercellular	[40]
14	*Thymus capitatus*	HSV-2	18.6 μg/mL	6.0	Intercellular	[40]
15	*Salvia desoleana*	Acyclovir-resistant HSV-2	28.6 μg/mL	55.2	Intercellular and intracellular ^a^	[2]
16	Mexican oregano (*Lippia graveolens*)	Acyclovir-resistant HSV-1	55.9 μg/mL	13.1	Intercellular	[33]
17	Patchouli	IFV-A (H1N1)	0.088 mg/mL	1.15	ND	[41]
18	*Cinnamomum zeylanicum*	IFV-A (H1N1)	<3.1 μL/mL	>4	Intercellular	[23]
19	*Citrus bergamia*	IFV-A (H1N1)	<3.1 μL/mL	>5	Intercellular	[23]
20	*Cymbopogon flexuosus*	IFV-A (H1N1)	<3.1 μL/mL	>4	ND	[23]
21	*Thymus vulgaris*	IFV-A (H1N1)	<3.1 μL/mL	>4	Intercellular	[23]
22	*Lavandula officinalis*	IFV-A (H1N1)	<3.1 μL/mL	>8	Intercellular	[23]
23	*Eucalyptus globulus*	IFV-A (H1N1)	<50 μL/mL	>0.5	Intercellular	[23]
24	*Pelargonium graveolens*	IFV-A (H1N1)	<3.1 μL/mL	>21	Intercellular	[23]
25	*Citrus reshni* ripe fruit peel	Avian influenza virus A (H5N1)	2.5 μg/mL	8.7	ND	[11]
26	*Fortunella margarita* fruit	Avian influenza virus A (H5N1)	6.8 μg/mL	ND	ND	[12]
27	*Thymus vulgaris vulgaris*, *Cymbopogon citratus*, *Rosmarinus officinalis*	HIV-1	0.05–0.83 μg/mL	1.13–3.6	ND	[25]
28	*Cymbopogon nardus*	HIV-1	1.2 mg/mL	ND	ND	[31]
29	*Dysphania ambrosioides*	Coxsackie virus B4	21.7 μg/mL	74.3	ND	[42]
30	*Osmunda regalis* (Tunisian fern)	Coxsackie virus B4	2.2 μg/mL	789.8	ND	[43]
31	Eucalyptus globulus bicostata	Coxsackie virus B3	0.7 mg/mL	22.8	Intercellular	[44]
32	Patchouli	Coxsackie virus B3	0.081 mg/mL	1.2	ND	[41]
33	Mexican oregano (*Lippia graveolens*)	Bovine viral diarrhoea virus	78 μg/mL	7.2	Intracellular	[33]
34	*Ocimum basilicum*	Bovine viral diarrhoea virus	474.3 μg/mL	3.7	ND	[45]
35	Patchouli	Respiratory syncytial virus	0.092 mg/mL	1.1	ND	[41]
36	Mexican oregano (*Lippia graveolens*)	Respiratory syncytial virus	68 μg/mL	10.8	Intercellular	[33]
37	*Lippia alba*	Yellow fever virus	4.3 μg/mL	30.6	Intercellular and intracellular	[46]
38	Patchouli	Adenovirus-3	0.084 mg/mL	1.2	ND	[38]
39	Mexican oregano (*Lippia graveolens*)	Bovine herpes virus 2	58.4 μg/mL	9.7	Intercellular and intracellular	[33]
40	*Ayapana triplinervis*	Zika virus	38 μg/mL	12.5	Intercellular	[47]
41	*Teucrium pseudochamaepitys*	Coxsackievirus B	589.6 μg/mL	1.11	ND	[48]

Note: SI, selectivity index; HSV, human herpes viruses; HIV, human immunodeficiency virus; ND, not determined. “Intercellular” signifies mechanisms of action targeting virus surface during or before adsorption, while “intracellular” indicates action on intracellular events in viral lifecycle. ^a^ The late stage of viral lifecycle.

**Table 2 molecules-25-02627-t002:** Antiviral activity of essential oil-derived components.

No	Components	Viruses	IC_50_	SI	Intercellular or Intracellular Mechanisms	References
1	β-Caryophyllene	HSV-1	0.25 μg/mL	140	Intercellular	[39]
2	Farnesol	HSV-1	3.5 μg/mL	11.4	Intercellular	[39]
3	β-Eudesmol	HSV-1	6 μg/mL	5.8	Intercellular	[39]
4	Trans-anethole	HSV-1	20 μg/mL	5	Intercellular	[39]
5	Eugenol	HSV-1	35 μg/mL	2.4	Intercellular	[39]
6	Thymol	HSV-1	0.002%	7	Intercellular	[14]
7	Carvacrol	HSV-1	0.037%	1.4	Intercellular	[14]
8	*p*-Cymene	HSV-1	>0.1%	ND	Intercellular	[14]
9	Carvacrol	HSV-1	48.6 μg/mL	5.1	Intracellular	[33]
10	Thymol	HSV-1	7 µM	43	Intercellular	[49]
11	Carvacrol	HSV-1	7 µM	43	Intercellular	[49]
12	β-Pinene	HSV-1	3.5 μg/mL	24.3	Intercellular	[50]
13	Limonene	HSV-1	5.9 μg/mL	10.2	Intercellular	[50]
14	Carvacrol	Acyclovir-resistant HSV-1	28.6 μg/mL	8.7	Intracellular	[33]
15	Carvacrol	IFV-A (H1N1)	2.6 μg/mL	<0.15	ND	[51]
16	Eugenol	IFV-A (H1N1)	<3.1 μL/mL	ND	ND	[23]
17	β-Santalol	IFV-A (H3N2)	10–100 μg/mL		Intracellular ^a^	[52]
18	Germacrone	IFV-A (H1N1)	6.03 μM	>41	Intercellular and intracellular ^b^	[53]
19	1, 8-Cineole	BVDV	331.17 μg/mL	9.1	Intercellular	[45]
20	Camphor	BVDV	318.51 μg/mL	13.9	Intercellular	[45]
21	Thymol	BVDV	248.56 μg/mL	5.6	Intercellular	[45]
22	Carvacrol	BVDV	50.7 μg/mL	4.2	Intracellular	[33]
23	Carvacrol	Bovine herpes virus 2	663 μg/mL	0.3	Intracellular	[33]
24	Carvacrol	Respiratory syncytial virus	62 μg/mL	4.1	Intracellular	[33]
25	Carvacrol	Human rotavirus	27.9 μg/mL	33	Intracellular	[33]
26	Citral	Yellow fever virus	17.6–25 μg/mL	1.1–1.5	ND	[46]
27	β-Caryophyllene	Dengue virus	22.5 μM	71.1	Intercellular and intracellular ^b^	[13]
28	Thymohydroquinone dimethyl ether	Zika virus	45 μg/mL	9.1	Intercellular	[47]

Note: SI, selectivity index; HSV, human herpes viruses; HIV, human immunodeficiency virus; BVDV (Bovine viral diarrhoea virus); ND, not determined. “Intercellular” signifies mechanisms of action targeting virus surface during or before adsorption, while “intracellular” indicates action on intracellular events in viral lifecycle. ^a^ The late stage of viral lifecycle. ^b^ The early stage of viral lifecycle.

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
