# Peer review of "Antiviral Effects of Plant-Derived Essential Oils and Their Components: An Updated Review"

_molecules, 2020, doi:10.3390/molecules25112627_

Round 1

Reviewer 1 Report

Dear authors,

The article must be revised by British English speaker for language improvement.

Some examples:

Lines 26-27 The infectious viral diseases remain a significant worldwide concern.

Viral diseases are infectious. Why not simply, viral infections or viral diseases?

Line 33 immunodeficiency virus (HIV) infection, interferes with, through its azido moiety, the formation of

“Azido moiety :or “azide moiety”

Line 49 “Satureja hotensis” or  “Satureja hortensis”

Lines 52-54 “Although there emerge a large quantify of studies probing the antiviral activities of plant derived EOs in the past decades, these studies are fairly extensive and not interconnected to form an overall trend or rules.” The structure of sentence is not correct.

Lines 54-56 ” The antimicrobial, antifungal, antioxidant and anti-inflammatory effects have been extensively reviewed at times [15-17], while the generalization of antiviral effects were seldom addressed and updated.” The structure of sentence is not correct.

Lines 61-63 “ In view of this, in this work, we focus on in vitro antiviral studies published within 10 years, with the purpose to provide up-to-date information on the antiviral properties of EOs.” Improve this sentence.

Lines 63-64 “Viruses with animal cells as the host were covered herein while phytoviruses are out the scope of the article.” Professional English native speaker should improve this sentence. Authors mean animal viruses.

Lines 79-80 “So far, the time-of-addition assay is the most widely used procedure in literature to investigate the overall intercellular and intracellular inhibitory properties of EOs.” Professional English native speaker should improve this sentence. Does the sentence need “in literature”.

Line 80-81 “Collectively, EOs and their components action directly, in a larger part, on free viruses (intercellular mode of action), while multiple modes of action should not be ruled out, which is case dependent.” Improve this sentence.

Line 91-92 “While 0.5% carvacrol treated murine norovirus grew from normal to ~900 nm in diameter, resulting in capsid disintegration.” Is it really the growth of virus?

Line 94-96 “Nevertheless, murine norovirus with capsid partially degraded may still be infectious since capsid degradation failed to lead to noticeable viral RNA reduction, as detected in a RNase I protection assay.” Review this sentence for improving.

Lines 100-102 “Hemagglutinin can cause hemagglutination of red blood cells, so  hemagglutination inhibition assay is usually used to test the effect of EOs on viral adsorption to host cells”. Use your abbreviation HA (line 99). HA cause agglutination of red blood cells. Hemagglutination is agglutination of red blood cells. Review this sentence for improving.

Lines 104-105 “Summing up, EOs most likely action on, if there is any, HA than on NA and it is EO or compound specific”. Review this sentence for improving.

Line 105 “Cedar leaf oil” or “cedar leaf EO”?

Line 106 “а-pinene” should be changed to ”α -pinene”.

Lines 110-113 “Feriotto et al. [24], by  examining EO treated Tat/TAR-RNA complex using gel electrophoresis, showed that EOs of Thymus vulgaris, Cymbopogon citratus, and Rosmarinus officinalis interacted directly with Tat protein and destabilized Tat/TAR-RNA complex.” Review this sentence for improving.

Line 114-117 “To investigate the antiviral activities of EO-derived components on Dengue virus, a non enveloped RNA virus, Pajaro-Castro et al. [13] calculated in silico the affinity scores of each component to Dengue virus proteins and then identified the interactions between the highly scored components and the viral protein, with the aid of software AutoDock Vina and LigandScout 3.0.” Review this sentence for improving.

Lines 119-121 “However, whether this hydrophobic interactions apply to cases with other viruses and EOs involved and will the polarity of the EO components affect the antiviral efficacy remain to be elucidated”. Review this sentence for improving. Singular, Plural, …

Lines 123-124 “To infect the host cells, the IFV needs to uncoat, which need an acidic endosomal and lysosomal environment”. Review this sentence for improving.

Line 126 “Tea tree oil” or “tea tree EO”?

Lines130-131 “Also, Eos may action on genome related sites as revealed by genetic approaches”. Is it a verb “action” the most appropriate one?

Lines 131-133 “Towards a certain type of virus, mechanisms of action may be essential oil dependent [29], possibly owing to the constitutional variations of the Eos”. Why not to use abbreviation EO?

Lines133-134 “Summing up, there is a tendency that EOs act directly on free viruses is the most common mode of action”. Review this sentence for improving.

Line 135 “3. The antiviral activities of essential oils” should be improved. The section should be connected.

Lines 140-142 “To ascertain that the assayed concentrations of EOs do not exert toxicity on the host cells, we must test the cytotoxicity of EOs, which is described in terms of CC50 (50% cytotoxic concentration), corresponding to the EO concentration that reduces the cell viability by half.” In my opinion English native speaker should review this sentence for improving.

Lines 142-143 “Antiviral selection index (SI) was calculated as the ratio of CC50 to IC50 ”

Line 143-144  “To define anti-infective potential in natural products, IC50-values of <100 μg/ml applies for mixtures and <25 μM for pure compounds”. In my opinion Review this sentence for improving.

Table 1

Abbreviation is used for HSVs. Abbreviation “IFV” should be used also.

Selection indices should be rounded for 8-12, 29- 30, 32, 34 plants.

“Tunisian Fern” shoul be changed to “Tunisian fern”.

Eucalyptus bicostata or Eucalyptus globulus bicostata?

Cymbopogon Nardus should be changed to Cymbopogon nardus.

Some IC50s could be rounded.

Lines 149-151, 154-156 Intercellular mechanisms of action referrer to targeting virus surface during or before adsorption, while intracellular mechanisms of action refer to the action on intracellular events in viral lifecycle. Review this sentence for improving.

Yellow Fever Virus should be changed to Yellow fever virus.

Table 2

The first/second letter for the compotents? Capital or low one?

Use abbreviations for viruses.

Round some values of ic50 and SI.

Use abbreviations for “the early and late stage of viral lifecycle”. (Table 1 too)

Line 164 Use Italic for Umbelliferae, Labiatae, Myrtaceae, Lamiaceae

Check β-caryophyllene or β-Caryophyllene.

“β-Caryophyllene-bearing plants” – bearing?

Lines 169-171 “In a study, a widely encountered components in EOs have been screened for their anti-herpes activities and all exhibited high antiviral activities at the concentration range of 0.025-0.8 μg/ml with the exception of L-bornyl acetate and D-limonene”. The sentence should be improved.

Line 175 Check “Influenza viruses are enveloped RNA virus”.

Lines  176-178 “H.- J. Choi [38] screened the anti-influenza A/WS/33 virus activities of 62 EOs and found marjoram, clary sage and anise oils thereof exhibited higher efficacy (IC50 <100 μg/ml) than oseltamivir.” Check the grammar, language.

Lines 182-184 “Germacrone was also evidenced to effectively inhibit against multiple strains of feline caliciviruses, non-enveloped RNA viruses.” Check the grammar, language.

Line190 “This contrasting results

Lines 191-193 “There seems a trend that non-oxygenated terpene hydrocarbons are more effective to HSV and oxygenated terpenes to IFV, a hypothesis need to be  tested”. Improve grammar, language.

Lines 195-196 “ Antiviral studies in the past years focuses on enveloped viruses, while little work has been performed on non-enveloped viruses”. Improve grammar, language.

Lines 198 “While a non-enveloped virus reduces number of possible targets on which the EOs action.” Improve grammar, language.

Lines 203-204 “These studies evidenced the great potential of EOs on inactivating non-enveloped viruses.”. Improve language.

Line 208 “Cymbopogon Nardus” should be changed to “Cymbopogon nardus

Line 212 “However, the SI value is all below the recommended value of 4.” Improve language.

Line 223 Glechon Spathulata  should be changed to Glechon spathulate.

Line 229, 231 а-terpinyl acetate or α-terpinyl acetate?

Line 233, 234 The values “in μg/ml”could be rounded.

Line 236 “essential EO-derived phenylpropanoids”?

Lines 243-245 “Since no further research is  conducted to compared the antiviral effect of monoterpenes and sesquiter[emes, it is arbitrary to  draw a proposition that the antiviral competency is ascribed to the monoterpenes. ” check the grammar, language.

Line 247 Dengue protein?

Line 249 a should be changed to α

Line 256 two characteristic “component” should ne changed to “components”

Line 285 EOs instead of EO.

Line 292 “the EO isolates” ?

Line 299 “oseltanivir”

Line 301,302 “osletamivir”?

Line 311 “ailment” is not used in Medicine or Veterinary.

Lines 315-316 “Many  studies evidenced

Lines 322 “virial”

Line 327 “modeling”

Lines 328-335 “Additionally, it is known that the EO chemistry determines its bioactivity, efforts need to be put into probing the interplay between EO chemistry and its bioactivity. As discussed in the previous section, the antiviral effectiveness of EOs can be ascribed to the active components, either minor or principle, and underlying synergism, and the contribution of each component to overall antiviral efficacy may not be equal. Considering that many EOs share common constituents, there arises a problem that many EOs are possibly more or less bioactive to one virus type, and it is the same with one EO versus multiple viruses. In this case, screening of potent antivirals with vast plant resources is rather time-consuming and arduous.”. Every sentence should  be corrected.

Line 340-341 “The advent of advanced modern equipment facilitates the  fractionation of EO components and evaluation of their bioactivities.” ?

Lines 345-347 “Last but not least important, compared with ten years ago, the number of references regarding the antiviral effect of EOs does not rise too much and the focus of the virus type are still enveloped ones, with HSV-1 and IFV the most concerned.”Grammar, language.

Reviewer 2 Report

Review of Ma and Lei, 2020

*Overall, the quality of the English language is below the standards for the journal Molecules. The authors should seek the assistance of a native English speaking person to lift the quality to an acceptable level.

*The need for a review like this is intrinsically questionable, since the authors recognize themselves that the stage of understanding the mechanisms up to the level of biomolecular interactions could not be reached in the last decade. Giving that serious limitation, it is also not particularly clear why a focus on essential oils from plant origin is absolutely a priority, as compared to the importance of other phytochemicals as antiviral agents for which information at the level of biomolecular interactions is emerging (e.g. Mothashi et al., 2001; Chandani et al., 2019).

Reviewer 3 Report

Dear colleagues, dear editorial board,

the present manuscript of Ma and Lei reviewing antiviral effects of plant-derived
essential oils should be considered for publication. It is a well written review and
details about current experimental work in the field.
Maybe it is of up-to-date interest, to cite an in-silico study of the anti-COVID19 action of
essential oils (for example at l 327; Abdelhi et al, doi: 10.1080/07391102.2020.1763199). Furthermore, I would like to suggest citations of clinical studies
(Duijker et al., doi:10.1016/j.jep.2015.01.030, Nikakhtar et al. doi: 10.1002/ptr.6131)

#1 l 50: citation for the statement is missing.
#2 l 63 check: ....viruses with...Suggestion: ..viruses infecting..
#3 l 120 ..will... Suggestion: ..whether...
#4 ll 191-193 Suggestion: There seems to be a trend a hypothesis, which needs to be tested
#5 l 197 Suggestion: …,while
#6 section 4 should be integrated as 2.4/2.5?
#7 l 266 Suggestion ...while the later has not...
#8 l 279 please check ...used whole (?)...

I hope, the comments are of any help.

Best regards

Round 2

Reviewer 2 Report

The manuscript has been improved by taking into account my concerns. However, it remains weird that the abstract claims that underlying mechanisms will be explained, while in the conclusion and rebuttal it is stated that "essential oli-derived constituents were inadequately used for elucidation of the molecular level mechanism of virucidal activity"

Author Response

We do appreciate the reviewer's further comment. Mechanisms of the antiviral effect of essential oils and their constituents were mostly evaluated by investigators employing the time-of-addition experiment, which is not a good approach to track the molecular-level mode of action. To make it consistent, we changed "their mechanisms of action" in the abstract to "their overall mechanisms of action" (Line 17). And we added "at the molecular-level" into the last sentence but one (Line 352). Hopefully, our modification will make the meaning more unambiguous.